



# Weakly coupled atmospheric-ocean data assimilation in the Canadian global prediction system. (v1)

Sergey Skachko[1], Mark Buehner[1], Stéphane Laroche[1], Ervig Lapalme[3], Gregory Smith[2], François Roy[2], Dorina Surcel-Colan[3], Jean-Marc Bélanger[2], and Louis Garand[1]

[1]Data Assimilation and Satellite Meteorology Section, Meteorological Research Division, Environment and Climatic Change Canada, Dorval, Québec, Canada
[2]Environmental Numerical Prediction Research Section, Meteorological Research Division, Environment and Climatic Change Canada, Dorval, Québec, Canada
[3]National Prediction Development Division, Meteorological Service of Canada, Environment and Climate Change Canada, Dorval, Quebec, Canada

**Correspondence:** Sergey Skachko (Sergey.Skachko@canada.ca)

**Abstract.** A fully coupled atmosphere-ocean-ice model has been used to produce global weather forecasts at Environment and Climate Change Canada (ECCC) since November 2017. Currently, the system relies on four uncoupled data assimilation (DA) components for initializing the fully coupled global atmosphere-ocean-ice forecast model: atmosphere, ocean, sea ice and sea surface temperature (SST). The goal of the present study is to implement a weakly coupled data assimilation (WCDA) between the atmosphere and the ocean components and evaluate its performance against uncoupled DA. The WCDA system uses coupled atmosphere-ocean-ice short-term forecasts as background states for the atmospheric and the ocean DA components that independently compute atmospheric and ocean analyses. This system leads to better agreement between the coupled atmosphere-ocean analyses and the coupled atmosphere-ocean-ice forecasts than between the uncoupled analyses and the coupled forecasts. The use of WCDA improves the atmospheric forecast score near the surface, but a slight increase in the atmospheric temperature bias is observed. A small positive impact from using the short-term SST forecast on the satellite radiance Observation-minus-Forecast statistics is noted. Ocean temperature and salinity forecasts are also improved near the surface. Next steps toward stronger DA coupling are highlighted.

## 1 Introduction

Until recently, separate systems for atmospheric and ocean prediction have been used in the generation of operational forecast products at Environment and Climate Change Canada (ECCC). The numerical weather prediction (NWP) system used



a prescribed sea surface temperature (SST) while the ocean prediction system used the atmospheric forcing from the NWP system. However, it is established that coupled models can produce improved forecasts on various time scales (Neelin et al., 1994). At this time, several operational centers use coupled models to generate forecasts (e.g. ECCC, Smith et al. 2016; Met Office, Williams et al. 2018). At ECCC, the fully coupled atmosphere-ocean-ice model has been used operationally to produce

weather forecasts since November 2017.

Since coupled models have been shown to provide significant improvements versus uncoupled models for NWP systems, research leading to the implementation of various coupled data assimilation (CDA) strategies has started to potentially further improve the forecast skill. A review of current activities on coupled prediction systems and CDA may be found in Brassington et al. (2015). The World Meteorological Organisation (WMO) meeting on CDA (Penny and Hamill, 2017) defined the

classification of weakly and strongly coupled data assimilation as well as their variations. In this article, we follow the WMO definitions of CDA-related terminology.

Many CDA studies have already considered the coupled forecast initialization on timescales from seasonal to decadal (e.g. the Japan Agency for Marine-Earth Science and Technology (JAMSTEC), Sugiura et al. 2008; Mochizuki et al. 2016; the National Oceanic and Atmospheric Administration Geophysical Fluid Dynamics Laboratory (NOAA/GFDL), Yang et al. 2013;

Zhang et al. 2014). At the Met Office, a weakly coupled data assimilation (WCDA) system has been developed (Lea et al., 2015) to improve the forecast skill from short range to seasonal time scales, though this system is not yet operational. It is based on using a coupled atmosphere-land-ocean-ice model to compute the background states for separate atmosphere and ocean analyses in a 6-h assimilation window. Generally speaking, WCDA at the Met Office performed reasonably well providing results very similar to the uncoupled DA. In that study, the authors identified two main problems in their implementation of

WCDA: the ocean SST diurnal cycle issue and an erroneous coupled river runoff. The latter issue led to the degradation in salinity fields around some river basins. These results are nevertheless encouraging considering that the CDA system is new and neither the atmospheric nor ocean data assimilation systems were adjusted as part of implementing CDA.

A CDA system was developed at the European Centre of the Medium-Range Weather Forecast (ECMWF) (Laloyaux et al., 2016) to be used for the global coupled reanalysis of the recent climate. Their system consists of quasi strongly coupled atmo-

spheric and ocean DA using the coupled atmosphere-ocean model to compute updated short-term coupled forecasts during each outer-loop iteration for the 4D-Var atmospheric and 3D-Var ocean DA systems. They performed realistic CDA experiments and compared them with an uncoupled system. The results of CDA were similar to the uncoupled DA, with a small positive impact on the ocean temperature and slightly improved atmospheric temperature near the surface, especially in the Tropics. Ten day forecast skill scores for coupled atmospheric-ocean forecasts were mostly neutral with a small improvement in the Eastern

Tropical Pacific. The authors concluded that such a coupled system was a promising tool for investigating CDA methodology. They also pointed out further potential improvements of the CDA system. First, they stated that direct assimilation of SST and sea-ice observations may improve the use of these data as well as that of other near-surface observations. Second, coupled background error statistics with realistic representations of covariances between the atmosphere and ocean are needed.

To explore the later issue, Laloyaux et al. (2018) examined aspects of strongly coupled DA by using explicit cross-correlations

between the atmosphere and ocean background error estimated from an ensemble of coupled ocean-atmosphere models. In





comparison to their system with the coupled outer-loop, the use of explicit cross-correlations provides similar results. In addition, the authors estimated that when a fully-coupled ocean-atmosphere model is used in the outer-loop, 6 to 12 h of model integration are needed to synchronize the uncoupled ocean and atmospheric increments from the inner-loops. The authors pointed out that a shorter time, around 6 h, is needed to synchronize ocean increments in the regions where the cross-correlations are

large, including the Tropical and Northern Tropical Pacific and shallow mixed layer regions. The required time is however longer, around 12 h, in the midlatitudes or where the mixed layer is deep. The shorter synchronization time may potentially play a positive role in CDA systems with comparable DA windows introducing less initialization shocks into DA.

Storto et al. (2018) also developed a strongly coupled DA system using linearized atmosphere-ocean balance operators in a simplified framework. This work showed a positive impact on forecast skill, especially in the Tropics. Recently, Browne

et al. (2019) developed another CDA implementation to be used for the purposes of NWP at ECMWF. They showed that an NWP system may be degraded by model biases in the ocean component of the coupled atmosphere-ocean model. This is why they have chosen a weaker form of CDA than the system used for the coupled reanalysis. The chosen method was similar to WCDA, where the atmosphere and the ocean are coupled implicitly at a frequency of 24 h. The interaction between the two components is not performed using a coupled model but by using the analysis in one component to specify boundary

conditions in another component for the next 24-h cycle. This system resulted in smaller errors of atmospheric temperature and humidity in the regions from the surface up to 700 hPa. The WCDA also resulted in smaller errors for SST in the Tropics as compared to uncoupled analyses. However, the analysis increments within WCDA were significantly smaller than those within the uncoupled system in the Northern Extratropics, while in the Southern Extratropics the two systems behaved very similarly.

At ECCC, a WCDA between the atmosphere and land surface systems has been running for many years (Bélair et al., 2003),

whereas the ocean-ice and atmospheric components have remained uncoupled. The present study reports the development of a WCDA prototype between the atmospheric and ocean DA components. The chosen first prototype of WCDA at ECCC employs the same fully coupled atmosphere-ocean-ice model that is used operationally for medium-range forecasts, to compute coupled background states for the atmosphere and ocean DA systems. The aim of this approach is to investigate the impact of the evolving ocean temperature on the atmospheric DA using existing independent DA components and with no additional

tuning of the models. The new WCDA system is assessed using realistic experiments based on systems very similar to those used operationally at ECCC. The results of the new WCDA system are compared with the current uncoupled DA system.

The next section describes the individual DA and forecast components used in this study. The ways these individual systems are linked together within the complete forecast-analysis cycle are described for the uncoupled and WCDA approaches in Sect. 3.1 and 3.2, respectively. In Sect. 4, results from DA experiments are presented and analyzed. Conclusions are given in

Sect. 5.

## 2  Description of ECCC's atmospheric and ocean prediction components

In this section, we describe the DA schemes for the atmosphere, ocean, SST and sea-ice followed by a description of the fully coupled NWP forecast model used at ECCC. All DA components as well as the NWP models, coupled and uncoupled, have



already been validated and reported in numerous studies. Here we briefly describe only the parameters related to the present study.

## 2.1 Atmospheric data assimilation

The atmospheric DA component used in this study is the same as the Global Deterministic Prediction System (GDPS: Buehner et al., 2015; Charron et al., 2012) developed and used operationally at ECCC. The computation of the background state during the analysis cycle is performed using the Global Environmental Multiscale (GEM) atmospheric model (Côté et al., 1998a, b; Zadra et al., 2014) with a horizontal grid spacing of 25 km and 80 vertical terrain-following levels with the model top at 0.1 hPa. The turbulent surface heat and momentum fluxes are parametrized using stability functions described in Delage and Girard (1992) and Delage (1997).

The DA method is a hybrid four-dimensional ensemble-variational (4D-EnVar) (Buehner et al., 2013, 2015) using Incremental Analysis Update (IAU) initialization (Bloom et al., 1996). The IAU implementation used with the GDPS system is illustrated in Fig. 1. The hybrid approach combines four-dimensional ensemble covariances with the static error covariances computed with so-called NMC method (Parrish and Derber, 1992) to estimate the full spatio-temporal background error covariances over the 6-h assimilation time window. The ensemble and static covariances are averaged with equal weights in the resulting full background error covariances. The ensemble covariances are estimated from the ensemble of 256 background states, available hourly within the 6-h assimilation window, obtained from the global ensemble Kalman filter (EnKF) being used operationally at ECCC (Houtekamer et al., 2014) since 2005. The EnKF version used in this study (4.1.1) is the one that was operational at the time of the experiments. This version was replaced by an updated version in September 2018. The 4D-EnVar analysis increments are computed on a grid with a horizontal grid spacing of 50 km, as in the EnKF system.

In the current version of the GDPS, the operationally assimilated data consists of those from microwave and infrared satellite sounders and imagers, scatterometers, radiosondes, aircrafts, wind profilers, land stations, near-surface observations from ships and buoys, atmospheric motion vectors, ground-based GPS sensors and satellite-based radio occultation instruments. More details about the assimilated data may be found in Buehner et al. (2015).

The GEM model has been designed to be coupled with a land surface model; it is called ISBA: the Interactions between Soil–Biosphere–Atmosphere scheme (Noilhan and Planton, 1989). ISBA computes the surface heat and momentum fluxes over four surface types: land, glacier, water and sea ice. The prognostic surface variables of the current surface model are computed for the land surface only, which has its own DA system (Bélair et al., 2003). The land surface model and DA in both uncoupled and coupled DA experiments of this study are similar to what is currently used operationally. The definition of water and sea ice surfaces requires information from the external SST and sea-ice DA components, as described in Sect. 2.3 and 2.4, respectively.

## 2.2 Ocean data assimilation

The ocean DA component used in this study is the same as the current operational Global Ice-Ocean Prediction System (GIOPS: Smith et al., 2016). The numerical model used is the NEMO-CICE coupled ocean-ice model based on NEMO (Nucleus for





European Modelling of the Ocean) version 3.1.3 (Madec, 2008; Smith et al., 2018) and CICE (Community Ice CodE) version 4.0 (Hunke and Dukowicz, 1997; Hunke et al., 2010). The NEMO version used here has a global 1/4°horizontal grid and 50 vertical levels ranging from the ocean surface to the ocean bottom with spacing increasing from 1 m at the surface to 500 m near the bottom. The thermodynamic component of CICE computes growth and melt of snow and ice, and the vertical temperature

profile using four ice and one snow layers. The ocean upper boundary conditions are the turbulent surface latent, sensible, incoming and outgoing long-wave and short-wave radiation fluxes.

The ocean DA system used is the "Système d'Assimilation Mercator" version 2 (SAM2) (Lellouche et al., 2013). The SAM2 DA algorithm is the Singular Evolutive Extended Kalman (SEEK) filter derived from the Kalman filter (Pham et al., 1998). The background error covariances are estimated using an ensemble of multivariate three-dimensional anomalies derived from

a multi-year hindcast simulation (Lellouche et al., 2013). The version used in this study is GIOPS v2.2.3 corresponding to the currently operational system.

The SAM2 DA consists of two successive DA cycles. First, in a weekly cycle, in situ data, SST and satellite altimetry data are assimilated. The in situ data are derived from multiple sources including the Argo floats (Gould, 2005), as well as ships, moorings and instrumentation on sea mammals. The satellite altimetry data are obtained from a product combining anomalies

of the sea level with a mean dynamic topography. The sea level anomalies are provided by the Archiving, Validation and Interpretation of Satellite Oceanographic data (AVISO), Ssalto/Duacs near-real-time data that include Jason2, Cryosat2 and Saral/Altika satellite data. The mean dynamic topography used is from Rio et al. (2011). Second, in a daily assimilation cycle, only SST data are assimilated. The assimilated SST is itself a gridded analysis computed within the operational SST DA based on optimal interpolation (OI) as described in Sect. 2.3. Within this DA cycle, the sea-ice analysis, produced by the system

described in Sect. 2.4, is also used to reinitialize the model ice concentration each day.

### 2.3   Sea Surface Temperature data assimilation

The SST DA system is described in detail in Brasnett (2008); Brasnett and Colan (2016). The version used in this study is the operational SSTv1.2.2 which is currently employed in the operational implementation of the atmospheric GDPS system (see Sect. 2.1). The DA approach used in the system is the OI method producing daily analyses considered to be valid at 0000

UTC. The assimilated data are collected during the period of 24 h before this valid time. The SST OI assimilates data from multiple in situ platforms (drifting and moored buoys, ships) and Advanced Very High Resolution Radiometer data provided by the US Naval Oceanographic Office that are derived following the Multichannel SST (MCSST) approach (May et al., 1998; McClain et al., 1985). The SST DA system does not employ a numerical forecast model. Instead, it uses the previous analysis state computed 24 h earlier as the background state for the assimilation.

The assimilation of the SST data from multiple satellite instruments along with the in situ data is discussed in detail in Donlon et al. (2007). An important aspect of the system is the estimation and removal of observation biases. Indeed, the infrared radiometer measures the temperature within the conductive diffusion layer at a depth of ∼10-20 $\mu$m, or so called skin temperature. The microwave radiometer retrieves the sub-skin temperature at the base of the laminar layer at around 1 mm depth. All these data need to be reconciled with the in situ temperature usually measured at depth of ∼2 m. On the other hand,



the current weather prediction system requires an SST field being kept fixed through 24 h. To this end, the SST at a depth where the effect of the diurnal cycle is negligible was required. Since the in situ observations are located sufficiently deep, the data from different satellite instruments are adjusted to the in situ data first. Second, a so called foundation SST, the temperature at a depth without diurnal cycle, is estimated. The estimation is performed in two stages. First, the background state, which

is the anomaly field of the foundation SST, is computed by substracting from the analysis of a previous state a precomputed monthly mean climatology interpolated in time. After the analysis state of the foundation SST anomaly has been computed, the climatological field interpolated to the current day is added to it to obtain the SST analysis state. The output of this system is a daily analysis of the foundation SST field on an uniform $0.2° \times 0.2°$ latitude-longitude grid. This field is used within both the GDPS (Sect. 2.1) and GIOPS (Sect. 2.2) DA components.

## 10  2.4   Sea-ice data assimilation

The sea-ice concentration DA system is based on 3D-Var DA method (Buehner et al., 2016) implemented on a global domain at $\sim 10$ km resolution. As for the atmospheric DA component described in Sect. 2.1, the sea-ice analyses are computed every 6h at 0000, 0600, 1200 and 1800 UTC. For practical reasons, the sea-ice DA is designed without a forecast model but instead uses the previous analysis produced 6 h earlier as the background state for the assimilation. The 3D-Var sea-ice analysis assimilates

passive microwave satellite observations from the Special Sensor Microwave Imager (SSM/I), the Special Sensor Microwave Imager/Sounder (SSMIS) and Canadian Ice Service (CIS) manual analyses (Carrieres et al., 1996; Buehner et al., 2016). Details about how these datasets were treated within the 3D-Var DA may be found in Buehner et al. (2016). The sea-ice concentration analyses are used to initialize the atmospheric (see Sect. 2.1) and the ice-ocean (Sect. 2.2) forecast models to compute the background states for both uncoupled and coupled DA systems described in Sect. 3.

## 20  2.5   Coupled weather forecast model

In this section, the model used to produce medium-range forecasts in both, uncoupled and WCDA systems (see Sect. 3.1 and 3.2, respectively), is described. The system used in this study is the fully coupled atmosphere-ocean-ice GEM-NEMO-CICE model, that was operationally implemented at ECCC for global NWP in November 2017 (Smith et al., 2018). This system is used to produce 10-day coupled atmosphere-ocean-ice forecasts initiated at 0000 and 1200 UTC. The atmospheric

GEM model initial conditions are the corresponding 0000 and 1200 UTC atmospheric analyses (see Sect. 2.1) using the IAU initialization. The GEM model version used in this study is similar to the model described in Sect. 2.1 except for the modified ocean interface where the contributions from the ice-ocean model are transferred to the atmospheric model every model time step.

     The coupled NEMO version is 3.1.3, and CICE, 4.0, as described in Sect. 2.2. However, the models are launched in a coupled

mode that consists of a flux coupling approach described in detail in Roy et al. (2015) and Smith et al. (2018). This approach aims to simulate interactive and consistent transfer of heat, moisture and momentum between the atmospheric and ocean-ice models. It is based on common atmospheric, oceanic, and sea-ice state variables. First, the atmospheric model computes downward radiative fluxes and state variables and then transfers this information to the ocean-ice model. Second, these fluxes





and variables are used by the ocean model to compute turbulent upward surface atmospheric fluxes using the same formulation as used within the atmospheric GEM model (Delage and Girard, 1992; Delage, 1997). The turbulent surface fluxes over the ice covered areas are computed within the CICE model using stability functions described in Jordan et al. (1999). Finally, the surface fluxes are transferred to the atmospheric model along with the SST and sea-ice at every model time step of 15 min. The

time-stepping of the coupled atmosphere-ocean-ice model is implemented such that the atmospheric model moves forward one step ahead of the ocean-ice model prior to sending its variables.

The initial conditions for the NEMO ocean model are obtained from the daily SAM2 analyses at 0000 UTC (Sect. 22.2) for both forecasts launched at 0000 and 1200 UTC. The 0000 and 1200 UTC initial conditions for CICE ice model are obtained from the 3D-Var DA (Sect. 2.4) computed at 1800 UTC of the previous day, which assimilates normally more data than

analyses computed at 0000, 0600 and 1200 UTC.

## 3    Description of the complete forecast-analysis cycles

In the present section, the uncoupled DA (referred to UNCPL) and WCDA (referred to CPL) configurations of the complete forecast-analysis cycles are discussed. UNCPL is similar to the combined set of uncoupled systems currently used at ECCC for the operational production of forecasts. This system will be used in this study as a reference to evaluate the performance of

the new CPL system. Both systems use the DA and model components described in Sect. 2. The differences in design between UNCPL and CPL are in the assimilation cycles, whereas initial conditions from both UNCPL and CPL are used to initialize fully coupled atmosphere-ocean-ice 10 day forecasts (Sect. 2.5).

### 3.1    Uncoupled data assimilation

The graphical scheme of the uncoupled DA cycle is shown in Fig. 2. The atmospheric 4D-EnVar (Sect. 2.1) analyses are

computed four times per day at 0000, 0600, 1200 and 1800 UTC. The uncoupled atmospheric GEM model is initialized using IAU during 6-h integration centered at the analysis time followed by the 6-h GEM model integration to compute the background state for the next atmospheric DA (see Fig. 1). The GEM model forecast is computed using the SST and sea-ice fields specified by the separate DA systems described in Sect. 2.3 and 2.4, respectively. The GEM model version used in this system is the same as described in Sect. 2.1.

The weekly SAM2 ocean DA system of in-situ data and satellite altimetry is used in the same way in both UNCPL and CPL systems and not shown on the figure. The daily ocean SAM2 DA (Sect. 2.2) assimilating only SST data is computed at 0000 UTC. The ocean-ice NEMO-CICE model is initialized using IAU during 24 h model integration starting 24 h before the current analysis time followed by another 24-h NEMO-CICE model integration (see Sect. 2.2) to compute the background state for the next daily ocean DA. The ocean-ice NEMO-CICE model forecast during 24 h is run in uncoupled mode using the atmospheric

forcing fields from the GEM model forecast at 3-h frequency started at 0000 UTC.

The 3D-Var sea-ice DA (Sect. 2.4) computes analyses every 6 hours providing the ice concentration field used for the short-term forecasts used to produce the atmospheric background state. The 3D-Var sea-ice analysis computed at 1800 UTC





is used for initializing the 24-h ocean-ice forecast used as the background state for the following day. The SST OI analysis (Sect. 2.3) computed at 0000 UTC provides the static SST field used for the following four 6-h atmospheric forecasts used as the background states during 24 h. The same SST analysis is also assimilated into the SAM2 daily ocean DA system (see Sect. 2.2).

## 3.2 Weakly coupled data assimilation

The graphical scheme of the CPL cycle, combining coupled 6-h atmospheric and 24-h ocean DA systems, is shown in Fig. 3. The atmospheric 4D-EnVar analyses (Sect. 2.1) are computed every 6 hours (see Fig. 1) exactly as in the UNCPL system described in the previous section. However, the fully coupled atmosphere-ocean-ice model (see Sect. 2.5) is used within CPL to compute the coupled atmospheric background states for the atmospheric DA.

Only the atmospheric DA component of the CPL system uses explicitly the fully coupled atmospheric-ocean-ice model to compute the background states. To compute the observation-minus-background difference, the SAM2 ocean DA component directly uses the ocean-ice model launched in a forced mode, i.e. using precomputed atmospheric forcing fields. However, by saving the atmospheric fields from the 6-h coupled forecasts and using these to force the ocean model, this is equivalent to the explicit use of the fully coupled atmosphere-ocean-ice model. Preliminary experiments showed that the use of the precomputed atmospheric forcing from the fully coupled model every hour gives results similar to the forcing changing every model time step (the ocean-ice model time step is 15 min in our experiments).

Hence, in order to compute coupled 24-h background states for the ocean DA, the CPL system first performs four atmospheric 6-h DA cycles as shown in Fig. 1. The coupled atmospheric states from the coupled atmosphere-ocean-ice model integrations during the application of IAU are stored every one hour during this stage. To cover the whole 24-h cycle, the last three states are taken from the coupled background fields because the states during the use of IAU at 2200, 2300 and 0000 UTC are not yet available in the current 24-h DA cycle.

Once the entire 24-h period of atmospheric forcing (following the approach just described) is available, the ocean DA starts by integrating the ocean-ice model over the 24-h period to compute the observation-minus-background differences for SST. From these differences, the daily SAM2 ocean DA then computes analysis increments. The ocean analysis increment is then used to rerun the ocean-ice model over the same 24-h period using the same atmospheric forcing. This provides the initial conditions for the next 24-h cycle.

As in UNCPL, the 3D-Var sea-ice analysis (Sect. 2.4) is computed every 6 hours. However, only the analysis at 1800 UTC is used to initialize the computation of four 6 h fully coupled atmosphere-ocean forecasts used as background states during 24 h exactly as it is implemented in the the coupled weather forecast model (See Sect. 2.5). The SST OI analysis (Sect. 2.3) is computed at 0000 UTC and assimilated by the daily ocean SAM2 DA component as in the UNCPL system.





## 4  Comparison experiments

The experiments are conducted for the period of August-September 2017. The verification statistics shown in this section are computed between August $15^{th}$ and September $20^{th}$ during which a series of tropical Atlantic hurricanes occurred. Two 5-day forecasts per day at 0000 and 1200 UTC are carried out over this period. The atmospheric and ocean-ice initial conditions for

all experiments are taken from the DA systems described in Sect. 3.1 and 3.2. The comparison study focuses on the differences in the atmospheric and ocean forecasts and analyses produced by CPL and UNCPL systems. Differences between these two systems are expected for the SST as well as for near-surface layers in both atmosphere and ocean models.

The use of a coupled model between the atmosphere and the ocean-ice to compute the background state may lead to changes in both atmosphere and ocean DA. Concerning the atmosphere, these changes may be seen, for example, using satellite radi-

ances that are sensitive to the temperature of the ocean. Such instruments measure radiance within the atmospheric window, i.e. the sensitivity to the atmospheric temperature and humidity is low, which allows the emission from the ocean surface temperature to be measured. In order to illustrate the impact of the evolving ocean surface temperature on the atmospheric DA, the evolution of the Observation-minus-Analysis (OmA) and the Observation-minus-Forecast (OmF) biases and standard deviation are computed for the brightness temperature (TB) of the AQUA AIRS channel 950 (see Fig. 4 and 5, respectively).

6-h forecasts are used to compute the OmF biases and standard deviation. Generally, both OmA and OmF biases for this atmospheric window channel for UNCPL and CPL are similar. However, the CPL experiment results in smaller OmA and OmF standard deviations than UNCPL. Overall relative to UNCPL, the CPL standard deviations are reduced by 2.7% for OmA and 1.9% for OmF.

Such improvements in the OmF/OmA statistics for atmospheric window channel radiances may also be reflected in im-

provements of the medium-range atmospheric forecast. Figure 6 shows the difference in standard deviation of the forecasted atmospheric temperature against mean analysis (mean between the two experiments, CPL and UNCPL) for different pressure levels between 1000 and 10 hPa as a function of the forecast lead time. The red color shows the lead times and pressure levels where CPL performs better (i.e. has a lower standard deviation) than UNCPL, and the blue color, the converse. The score shown on the figure is computed in the Northern Extratropics region in the latitude band between 20°N and 60°N. In most

cases, CPL performs slightly better than UNCPL. A statistically significant difference with confidence above 90% is observed for the near-surface air temperature around 1000 hPa for the 12-h and 36-h forecasts. Similar forecast scores computed for the geopotential height and specific humidity also show slightly better performance of CPL, but with no areas where the improvement is statistically significant (not shown). The impact of CDA on forecast scores computed for the wind field is rather neutral (not shown). Similar results were also obtained when the ERA5 reanalysis is used instead of the mean analysis (also

not shown).

Figure 7 helps to highlight the better performance of CPL for forecast of near-surface air temperature over the same Northern Extratropics region. The top panel shows the standard deviation and bias of the air temperature at 1000 hPa against mean analysis as a function of the forecast lead time. The bottom panel shows the difference between the standard deviations of the





CPL and UNCPL. The CPL experiment results in standard deviation which is about 2% smaller than the corresponding value from the UNCPL experiment. However, CPL results in larger negative bias in the same area.

Let us now examine the quality of SST forecasts from the CPL and UNCPL experiments. Figure 8 shows the evolution of OmF standard deviation and bias for SST computed with respect to the original satellite and in situ observations that were used
within the SST DA system described in Sect. 2.3. The OmF statistics are computed for the 12 h coupled forecasts produced using the CPL and UNCPL initial conditions in three different latitude bands: Southern Extratropics [20°S, 60°S], Tropics [20°S, 20°N] and Northern Extratropics [20°N, 60°N]. The OmF standard deviations produced by CPL are systematically lower than those produced by UNCPL in all three regions. Whereas the OmF biases produced by CPL are sometimes larger.

Figure 9 shows the SST standard deviation and bias from the coupled medium-range forecasts produced using the CPL and
UNCPL initial conditions in the same three latitude bands as in previous figure. The standard deviation and bias are computed as a function of the forecast lead time with respect to the daily mean OSTIA product (Donlon et al., 2012) used in the ECMWF ERA5 reanalysis generated using Copernicus Climate Change Service information. This product was chosen to evaluate the forecasts from our experiments because it can be considered an independent dataset. Another high-quality SST product, the Group for High Resolution SST (GHRSST) Multi-Product Ensemble (GMPE) system (Martin et al., 2012), was not chosen
because it partially uses the same SST analyses from ECCC as described in Sect. 2.3. The standard deviations are similar for both experiments in all three latitude bands with slightly smaller values in the Tropics for CPL. The bias produced by CPL is smaller in the Tropics and Northern Extratropics, and slightly bigger in the Southern Extratropics.

Similar error statistics for SST can be computed within the ocean DA system described in Sect. 2.2. Figure 10 and 11 compare the SST OmF standard deviation and bias computed using 24-h forecasts within the CPL and UNCPL experiments.
The error statistics are computed relative to the gridded foundation SST field obtained from the SST DA that was assimilated by the daily SAM2 ocean DA as explained in Sect. 2.2. The use of the analysis field to compute the standard deviation and biases affects the spatial sampling as compared to OmF statistics against SST satellite and in situ data shown in Fig. 8. The figures show two typical OmF plots computed in the 'Puerto Rico XBT' region showing the performance in the Western Tropical and Northern Extratropical Atlantic ocean, and the 'Niño3' region, in the Tropical Pacific ocean. The CPL results in generally
smaller OmF biases and standard deviation errors in most regions.

The changes in the SST may modify the turbulent surface heat flux forecasts. To qualitatively evaluate the impact of CDA on the fluxes, the standard deviations for the turbulent surface sensible heat flux were computed using short 12-h coupled forecasts produced with the UNCPL and CPL initial conditions during September 2017. Figure 12 shows the difference between the two standard deviation fields. Positive values (red) mean that the standard deviation of the flux from the UNCPL experiment is
bigger than the standard deviation from the CPL experiment, whereas negative values (blue) mean the converse. The decrease in the standard deviations of surface sensible heat flux reflects a better accordance between the near-surface atmospheric temperature and the SST. In most regions, the impact of CDA on the surface sensible heat flux is neutral except the Northern Extratropical Atlantic where a series of tropical Atlantic hurricanes occurred as well as in the Northern Extratropical Pacific. In these regions, positive differences of 8-16 W/m2 and negative differences of 4-8 W/m2 are observed, though the number
of positive differences is bigger. Similar spatial structures but with smaller values are observed when the same quantities are



computed for the surface latent heat flux (not shown). It was noted during the evaluation of the fully coupled atmosphere-ocean-ice model (Smith et al., 2018) that interactions on the surface interface resulted in reduced latent heat flux due to the formation of cold wakes associated with cyclones leading to reduced intensification. Inclusion of the fully coupled atmosphere-ocean-ice model in the computation of background states improves the representation of these interactions in the analysis resulting in a

further decrease in the variance of fluxes (i.e. due likely to the reduced intensification of cyclones).

And finally, let us examine the impact of the WCDA on the subsurface ocean circulation. Fig. 13 and Fig. 14 show the difference between the root mean square errors (RMSE) for the monthly mean 24-h forecasts produced using UNCPL and CPL initial conditions relative to the Argo ocean temperature and salinity, respectively. The data used for the comparison are gridded fields using a global 1/2°horizontal grid and 58 vertical levels obtained from the "Monthly mean datasets of the

mean and annual cycle of temperature, salinity, and steric height in the global ocean from the Argo Program" (Roemmich and Gilson, 2009) downloaded from http://sio-argo.ucsd.edu/RG_Climatology.html. Positive values (red) indicate that RMSE of CPL is smaller, whereas negative values (blue) mean the converse. The upper plots in Fig. 13 and Fig. 14 show the RMSE differences for ocean temperature and salinity at 2.5 m depth, respectively. The lower plots show vertical sections of the differences between the RMSE from the UNCPL and CPL experiments. The vertical section for temperature is plotted through

the 1.5°N latitude between 120°W and 100°W. In most areas, CPL results in smaller temperature errors than UNCPL. The biggest differences for ocean temperature between the UNCPL and CPL experiments are observed in the Eastern Tropical Pacific, where the biggest positive differences of about 0.45°C form a front with the biggest negative values of about -0.5°C. Also for the vertical section, the errors produced by CPL are generally smaller in most areas between the ocean surface and 80 m depth. Below this depth, the difference between the CPL and UNCPL experiments are negligible. For ocean salinity (Fig. 14),

both CPL and UNCPL result in similar RMSE except in the Tropics, where CPL produces lower RMSE than UNCPL. The biggest values of the RMSE differences, around 0.55 psu, are observed in the Tropical Atlantic, where the vertical section of the RMSE differences for ocean salinity is shown through the 6.5°N latitude between 40°W and 15°W (bottom panel of Fig. 14). In this area CPL produces lower RMSE for the salinity between the surface and 20 m depth. The differences in salinity between UNCPL and CPL experiments are negligible below 70 m depth.

## 5 Conclusions

A WCDA system between the atmosphere and ocean DA components is implemented and evaluated in this study. The first prototype of the WCDA is built on the existing components of the NWP and operational ocean-ice prediction systems that have been previously run as independent uncoupled DA systems. As the NWP system requires SST and ice-concentration fields, and the quality of ocean-ice prediction depends on atmospheric forcing, the transition from uncoupled to strongly coupled DA

should be smooth and gradual in order to not degrade the quality of the existing atmospheric and ocean-ice prediction systems.

The first step towards CDA was to replace the uncoupled models used to compute the background state for each DA with the fully coupled atmosphere-ocean-ice model within separate atmosphere and ocean DA components. The ocean community model NEMO was already been coupled with atmospheric models to perform WCDA in multiple studies. However, the at-





mospheric GEM model and the 4D-EnVar DA were never tested before in a coupled framework. The present study showed that the use of the coupled atmosphere-ocean-ice model to compute the background states for separate atmospheric and ocean DA systems has generally neutral to positive effect on the 5-day atmospheric forecasts. However, the verification scores from WCDA are significantly better up to day 4 in the near-surface atmospheric layers in the Tropics and the Northern Extratropics.

WCDA also leads to better agreement between the near-surface atmospheric temperature and the SST decreasing locally the variability of turbulent surface heat fluxes as compared to uncoupled DA. Such encouraging results are obtained using the same configuration of the fully coupled atmosphere-ocean-ice model that was used operationally within the NWP system with no additional tuning of the model and allows one to explore further aspects of stronger coupling.

The improvement in the OmF standard deviation error for the near-surface air temperature is accompanied by a slight

increase of the bias for the same forecast lead times and pressure levels. It may be related to the current formulation of the error covariances that do not include cross-correlation terms between different DA components.

Another positive result is that the quality of ocean forecasts for SST and salinity with respect to both standard deviation errors and biases is improved when using the coupled background states for the ocean DA. This is noted by smaller verification errors for the ocean temperature and salinity, especially in the Tropics. This result is obtained despite the fact that the ocean

DA, as the atmospheric DA, does not employ the cross-correlation error covariances. Longer experiments are needed to further explore this issue.

In the current design of this first WCDA prototype, the daily ocean DA is mainly constrained by the uncoupled SST and ice-concentration analyses that do not rely on numerical models to compute background states. This weakens the degree of coupling because the daily initial condition for the whole atmosphere-ocean-ice system remains close to the uncoupled

trajectory and a certain time is needed to synchronize such initial conditions with the coupled model trajectory. The next step towards a stronger coupling would be to modify the SST and sea-ice concentration analyses such that they operate within a 6-h cycle and use coupled background states. The transition from a daily to the 6-h SST analysis will require a new bias correction scheme for the SST data in order to properly estimate the ocean temperature diurnal cycle. In addition, the purely technical step of using an integrated software for the atmosphere, SST and sea-ice analysis will be necessary as an initial step before

exploring stronger coupling by introducing background error cross-covariances between atmosphere and ocean components. This system may also be extended to perform the analysis for the whole ocean mixed layer, not only SST, every 6 hours. These 6-h analyses may replace the current daily ocean DA system. This is the general orientation of future work.

*Code and data availability.* The codes, scripts and data used in this paper are available for the Topical Editor and anonymous reviewers.

*Author contributions.* Concept, Sergey Skachko (S.S.), Mark Buehner (M.B.), Stéphane Laroche (S.L.), Gregory Smith; writing - original

draft preparation, S.S.; writing - review and editing, M.B., Louis Garand, S.L.; software, S.S., Ervig Lapalme, François Roy, Dorina Surcel-Colan, Jean-Marc Bélanger.



*Competing interests.* Hereby, we declare that no competing interests are present.

*Acknowledgements.* Authors would like to acknowledge Pierre Pellerin, Pierre Koclas, Nicolas Gasset, Jean-François Caron, Kristjan Onu, Mateusz Reszka, Kamel Chikar, Sylvain Heilliette, Charles Creese, Richard Ménard and many others for fruitful discussions and collaborative work on common software tools.





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



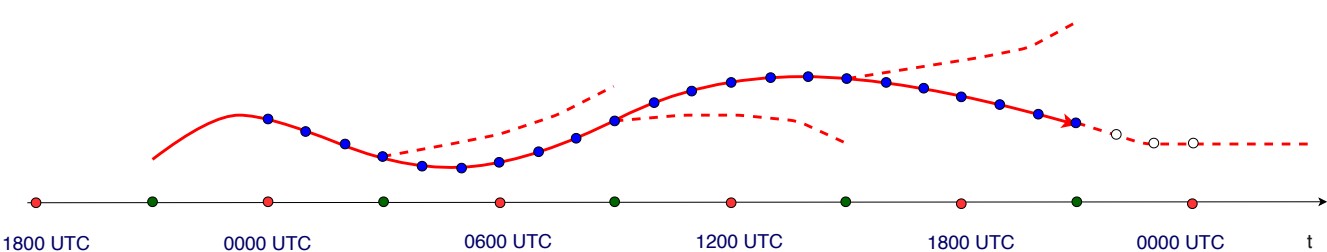

**Figure 1.** Graphical representation of 4D-EnVar analysis cycle during 24 hours using IAU initialization. The 4D-EnVar analyses are computed using 6 h assimilation window centered at 0000, 0600, 1200 and 1800 UTC (red dots on the time scale). The GEM model (in coupled or uncoupled mode) 6 h integration using IAU initialization of analysis increments (solid lines) starts at 2100, 0300, 0900 and 1500 UTC (green dots on the time scale), followed by 6 h computation of the background states for the next analysis (dotted lines). The 21 blue dots show the atmospheric states, available hourly, used as atmospheric forcing at 1 h frequency for the NEMO-CICE model to compute coupled 24 h ocean background within WCDA (see text). The last three forcing states of every 24 h cycle are taken from the computation of the background fields (white dots).

Zadra, A., McTaggart-Cowan, R., Vaillancourt, P. A., Roch, M., Bélair, S., and Leduc, A.-M.: Evaluation of Tropical Cyclones in the Canadian Global Modeling System: Sensitivity to Moist Process Parameterization, Monthly Weather Review, 142, 1197–1220, https://doi.org/10.1175/MWR-D-13-00124.1, https://doi.org/10.1175/MWR-D-13-00124.1, 2014.

Zhang, S., Chang, Y.-S., Yang, X., and Rosati, A.: Balanced and Coherent Climate Estimation by Combining Data with a Biased Coupled Model, Journal of Climate, 27, 1302–1314, https://doi.org/10.1175/JCLI-D-13-00260.1, 2014.



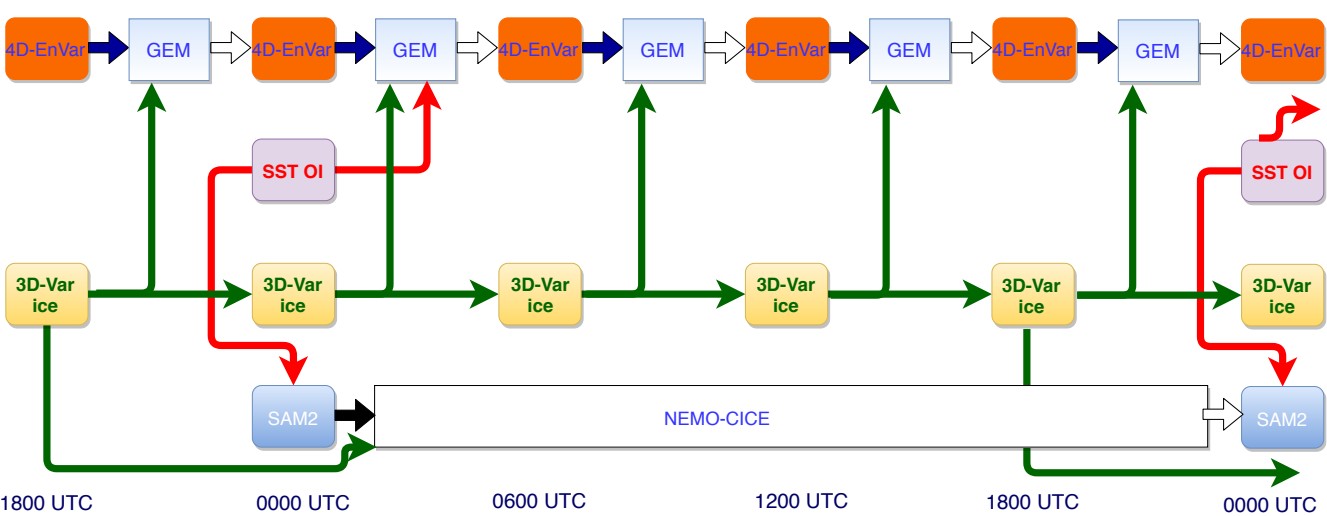

**Figure 2.** The uncoupled DA system (UNCPL) scheme. The atmospheric 4D-EnVar DA component computes analyses every 6 hours. The SST OI DA component computes daily analyses valid at 0000 UTC, that initialize the atmospheric GEM model and are assimilated using daily SAM2 ocean DA component at 0000 UTC. The 3D-Var ice analyses are computed every 6 hours and are used to initialize the uncoupled 6 h atmospheric and 24 h oceanic uncoupled forecasts. The uncoupled atmospheric analyses are propagated in space and time using the GEM model. The uncoupled ocean analyses are propagated using the NEMO-CICE model.

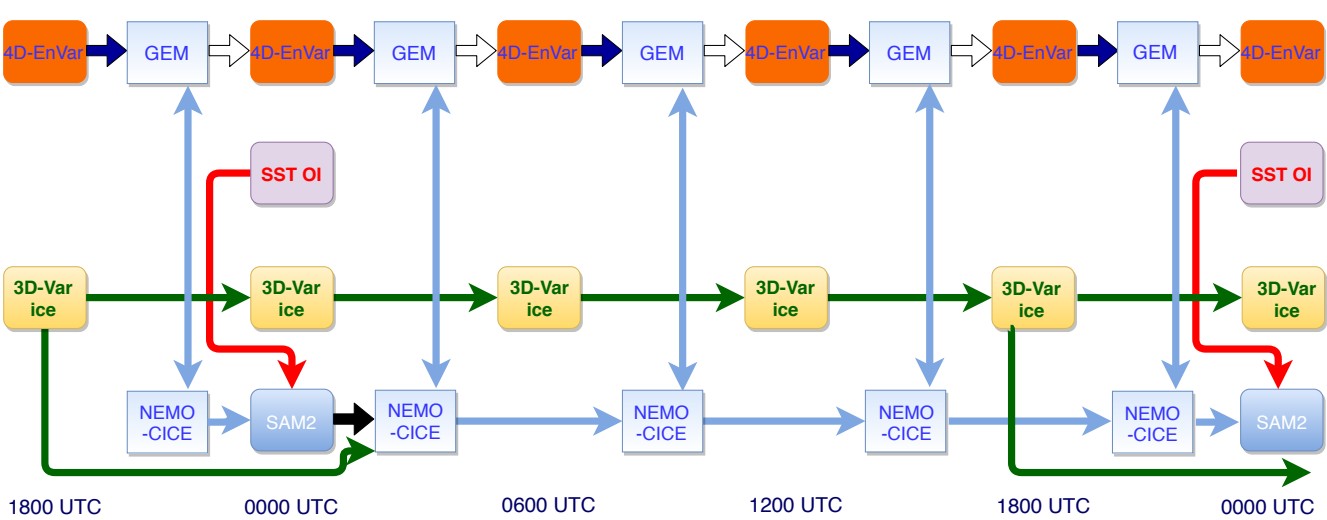

**Figure 3.** WCDA system (CPL) scheme. The atmospheric 4D-EnVar DA component computes analyses every 6 hours. The SST OI DA component computes daily analyses valid at 0000 UTC, that are assimilated using daily SAM2 ocean DA component at 0000 UTC. The 3D-Var sea-ice analyses are computed every 6 hours. The 3D-Var sea-ice analysis computed at 1800 UTC provides the initial condition for the computation of fully coupled atmosphere-ocean background states at 0000, 0600, 1200 and 1800 UTC. The separate atmospheric and ocean analyses are propagated in space and time using the fully coupled GEM-NEMO-CICE model.

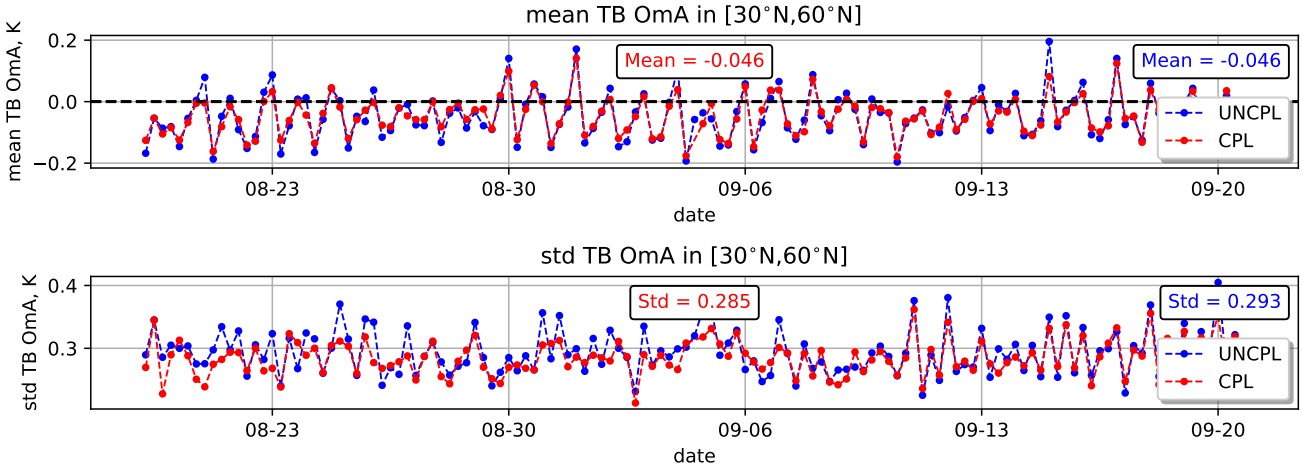

**Figure 4.** (a) Observation-minus-Analysis (OmA) statistics: bias (top) and standard deviation (bottom) using 6 h forecasts, statistics. Bias and standard deviation for the brightness temperature, in K, as seen by the AQUA AIRS Channel 950. The statistics are computed for UNCPL (blue) and CPL (red) in the Northern Extratropics region [30°N, 60°N].

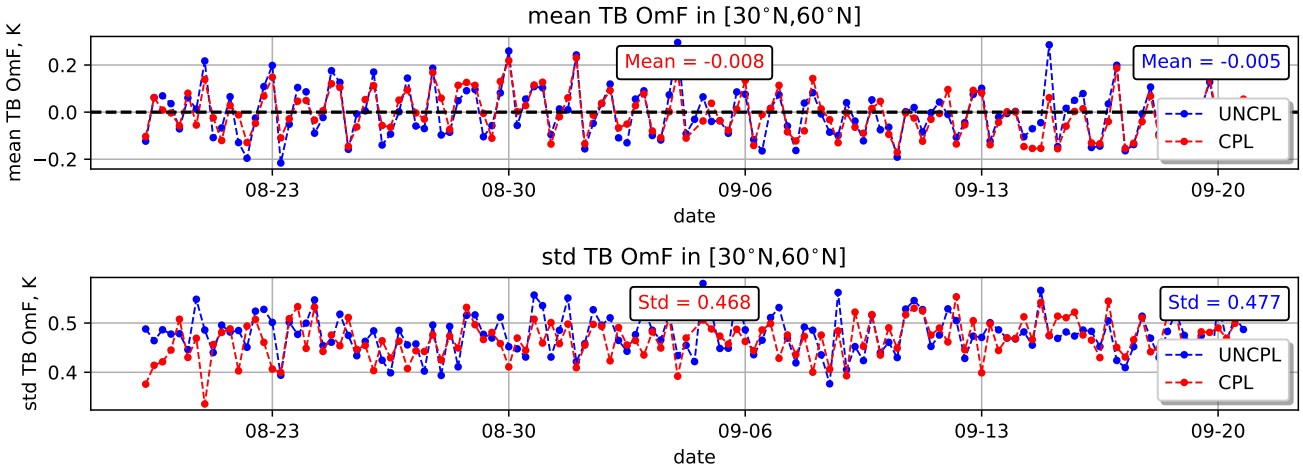

**Figure 5.** Observation-minus-Forecast (OmF) statistics: bias (top) and standard deviation (bottom) using 6 h forecasts, statistics. Bias and standard deviation for the brightness temperature, in K, as seen by the AQUA AIRS Channel 950. The statistics are computed for UNCPL (blue) and CPL (red) in the Northern Extratropics region [30°N, 60°N].

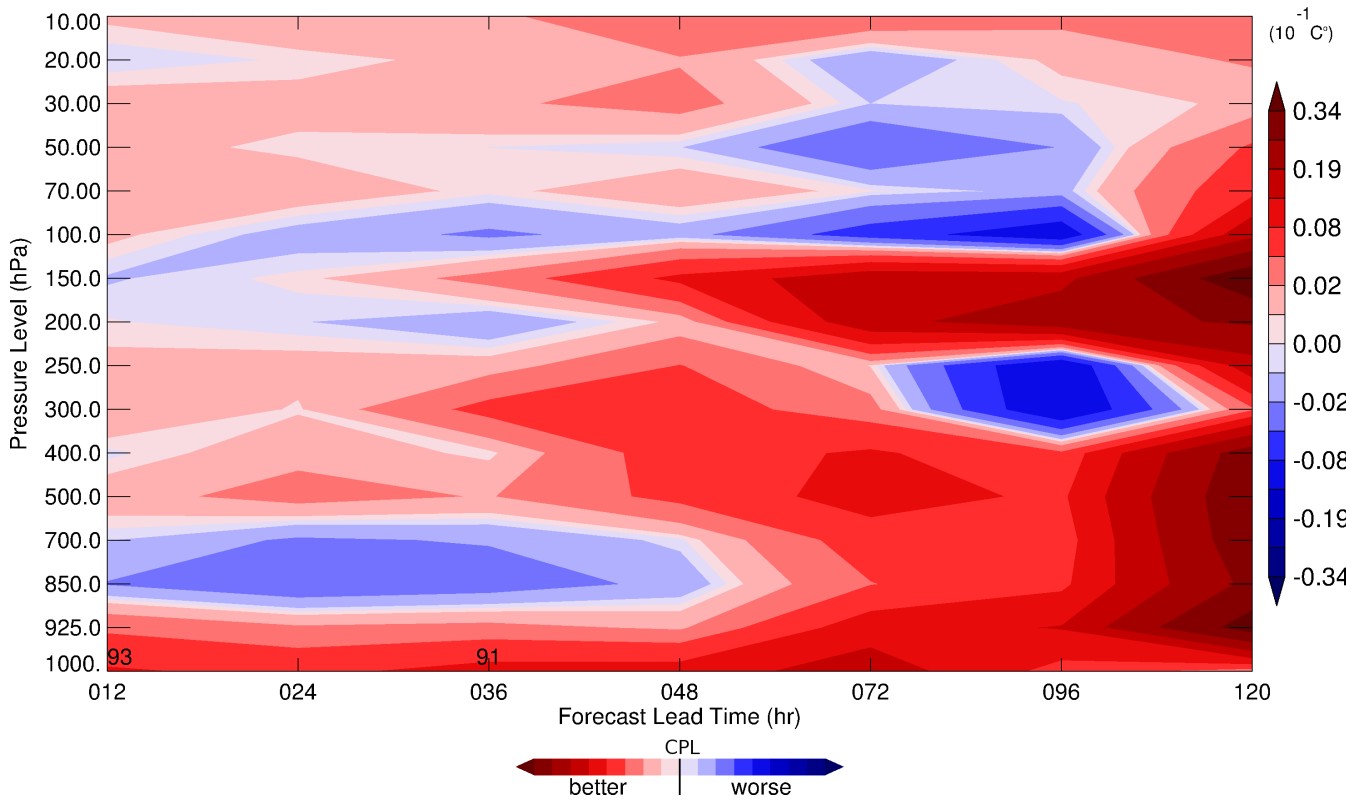

**Figure 6.** Difference in standard deviation of the air temperature, in degrees C, against mean analysis as a function of forecast lead time. The statistics are computed for CPL and UNCPL in the Northern Extratropics region [20°N, 60°N]. Positive values (red) mean that the standard deviation produced by CPL is smaller, whereas negative values (blue) mean the converse. Numbers show the areas where the difference between CPL and UNCPL is statistically significant with confidence level above 90%.



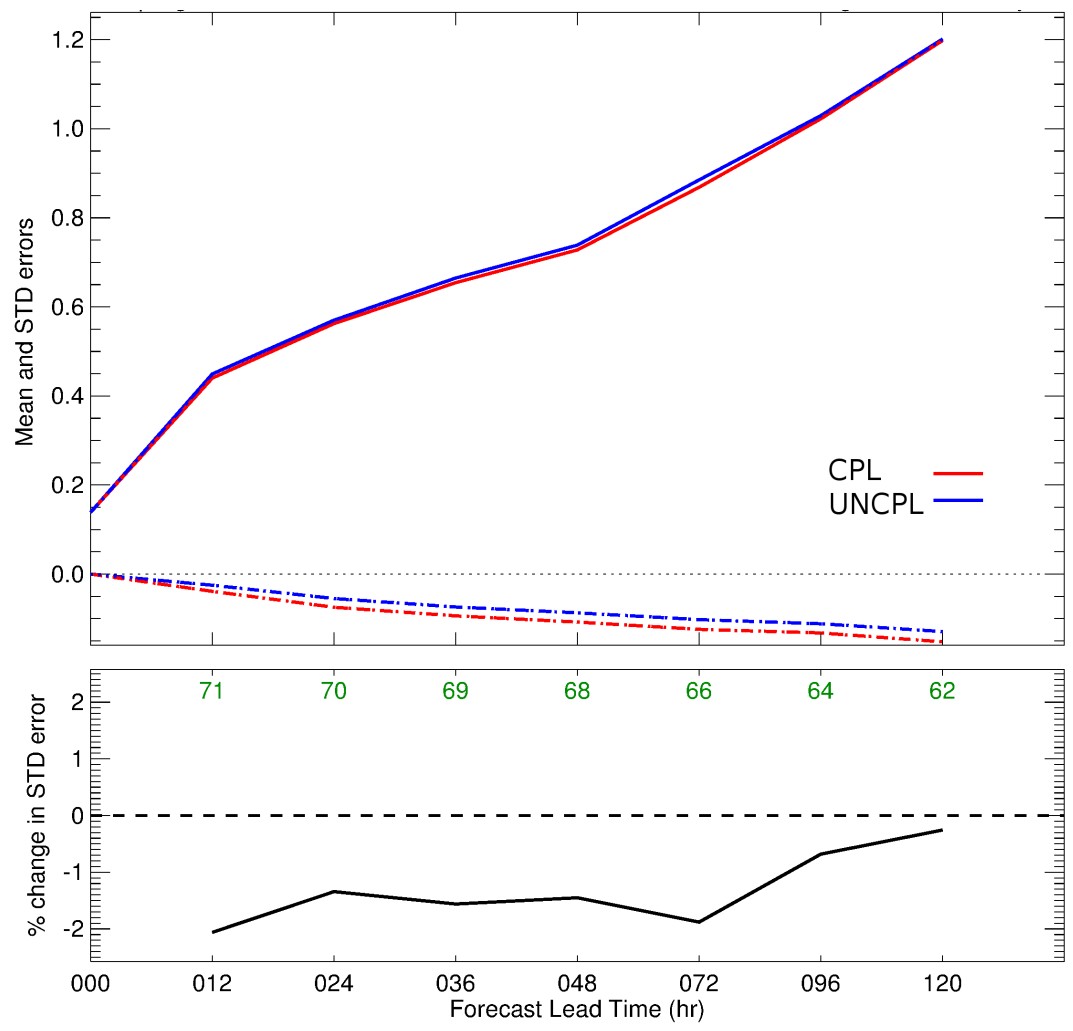

**Figure 7. Top panel**: standard deviation (solid curves) and bias (dashed curves) growth, in degrees C, as a function of forecast lead time for the air temperature at 1000 hPa. The statistics are computed for the CPL (red) and UNCPL (blue) against mean analysis in the Northern Extratropics region [20°N, 60°N]. **Bottom panel**: the change in the standard deviation error in % between CPL and UNCPL. The green numbers show the number of samples used to compute the statistics.

**Figure 8.** Evolution of sea surface temperature OmF standard deviation (top row) and bias (bottom row), in degrees C, with respect to SST data assimilated within the separate SST DA component. The statistics are computed for UNCPL (blue curves) and CPL (red curves) in three latitude bands: Southern Extratropics [20°S, 60°S] (left), Tropics [20°S, 20°N] (middle) and Northern Extratropics [20°N, 60°N] (right). The statistics are computed for the 12 h forecast and the corresponding data.



**Figure 9.** Standard deviation (top row) and bias (bottom row), in K, with respect to the ERA5 reanalysis daily mean SST fields as a function of forecast lead time. The statistics are computed for UNCPL (blue curves) and CPL (red curves) in three latitude bands: Southern Extratropics [20°S, 60°S] (left), Tropics [20°S, 20°N] (middle) and Northern Extratropics [20°N, 60°N] (right).

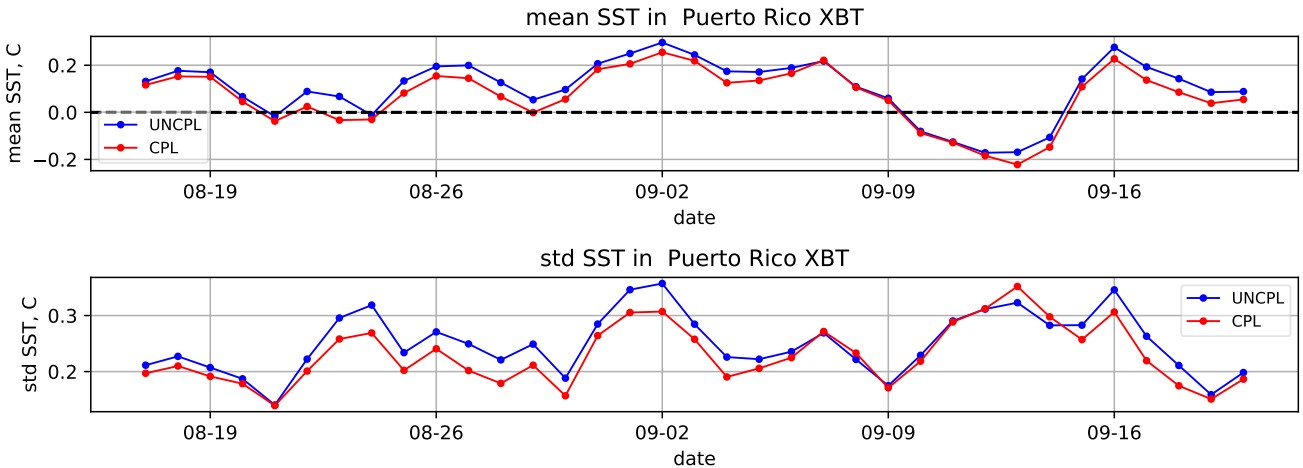

**Figure 10.** Bias (top) and standard deviation (bottom) with respect to the gridded foundation SST field from the ocean SAM2 DA component for the "Puerto Rico XBT" region situated within the latitude-longitude box defined by [35°W, 65°W] and [25°N, 35°N]. The statistics are computed using 24-h forecasts.

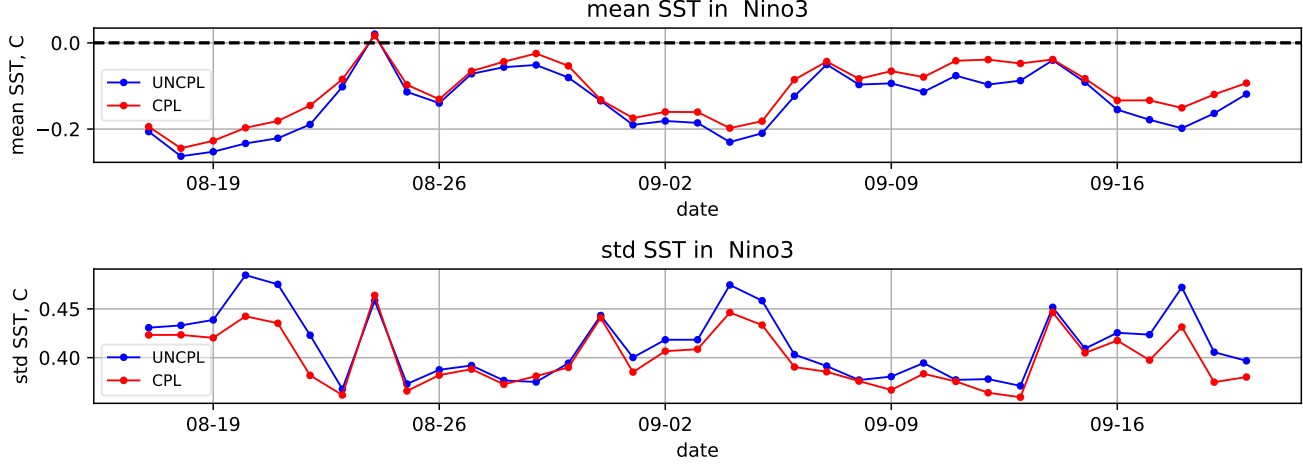

**Figure 11.** Bias (top) and standard deviation (bottom) with respect to the gridded foundation SST field from the ocean SAM2 DA component for the "Nino3" region situated within the latitude-longitude box defined by [150°W, 90°W] and [5°S, 5°N]. The statistics are computed using 24-h forecasts.



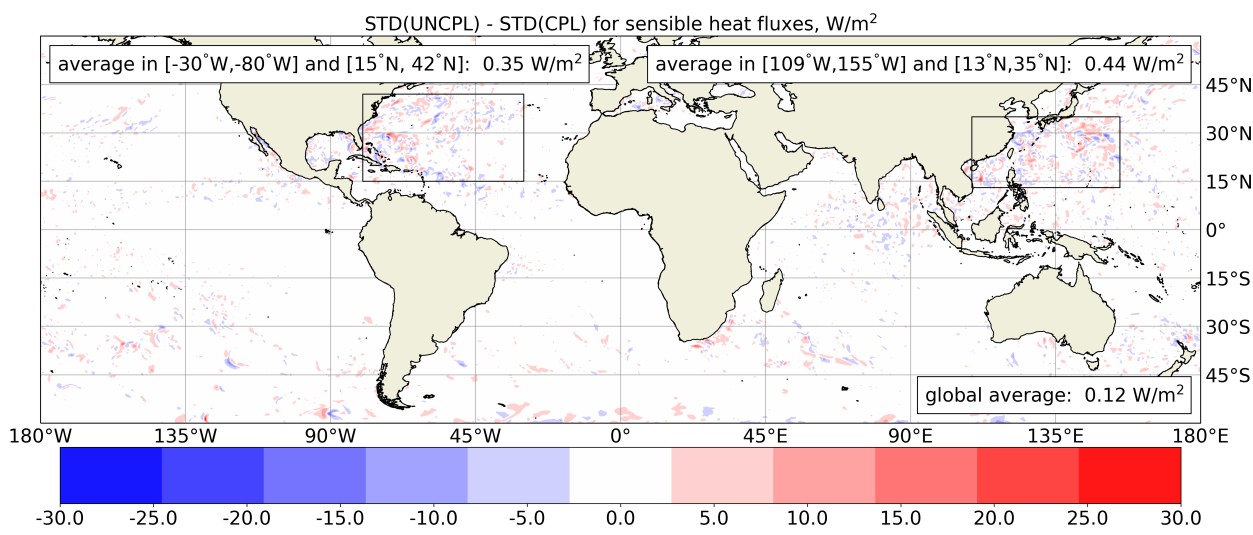

**Figure 12.** Difference, in W/m2, between standard deviation of UNCPL and CPL for the turbulent surface sensible heat flux computed using 12-h coupled forecasts during September 2017.



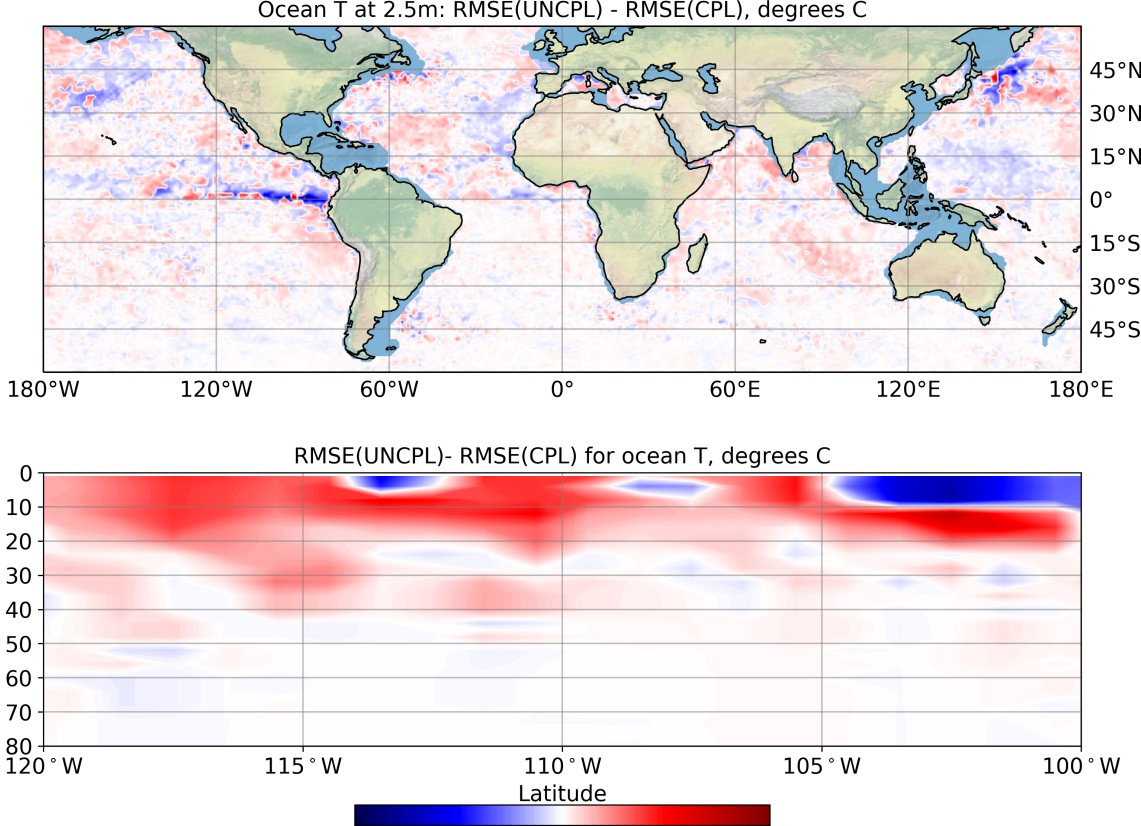

**Figure 13.** Difference between UNCPL and CPL (24-h forecasts) of the RMSE, in degrees C, with respect to the Argo ocean temperature measurements in September 2017. Positive values (red) indicate that the RMSE produced by CPL is smaller, whereas negative values (blue) mean the converse. The upper plot shows the RMSE difference for ocean temperature at 2.5 m depth. The grey areas show the regions where the Argo measurements are not taken. The lower plot shows the temperature vertical section through the 1.5°N latitude in the Eastern Tropical Pacific ocean between 120°W and 100°W.

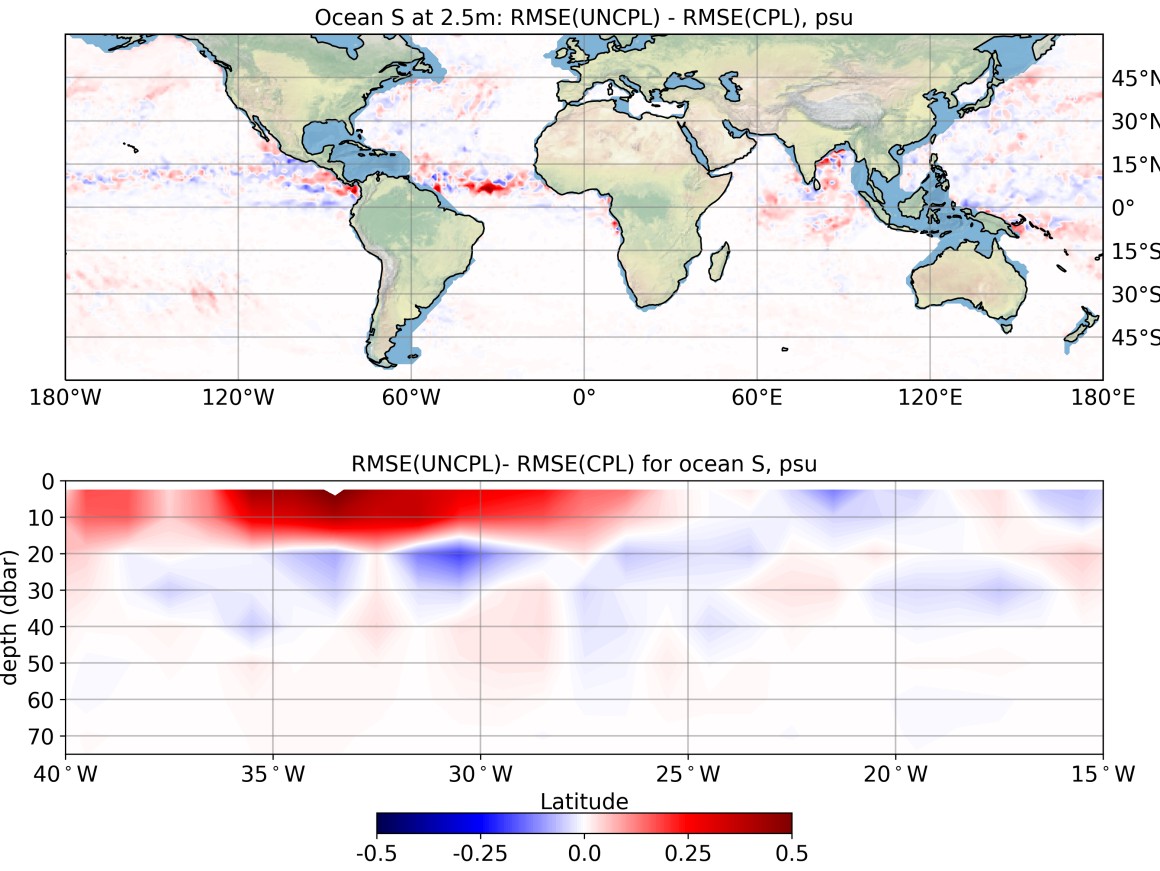

**Figure 14.** Difference between UNCPL and CPL (24-h forecasts) of the RMSE, in psu, with respect to the Argo ocean salinity measurements in September 2017. Positive values (red) indicate that the RMSE produced by CPL is smaller, whereas negative values (blue) mean the converse. The upper plot shows the RMSE difference for ocean salinity at 2.5 m depth. The grey areas show the regions where the Argo measurements are not taken. The lower plot shows the salinity vertical section through the 6.5°N latitude in the Tropical Atlantic ocean between 40°W and 15°W.