# Peer review of "Weakly coupled atmospheric-ocean data assimilation in the Canadian global prediction system. (v1)"

_Geoscientific Model Development, 2019_

## Referee Comment (RC1) · Anonymous Referee #1 · 13 Sep 2019

Overview

This paper describes a development to the Canadian global prediction system at ECCC to use weakly coupled ocean atmosphere data assimilation. The impact of this is assessed against a system using the previous uncoupled ocean and atmosphere DA approach. The paper is overall well written and it will be valuable to publish the work. I do have a main comment and some detailed comments which I believe should be addressed first listed below.

Main comment

The are some positive impacts from the coupled DA but I didn't feel I understood why

this was the case. You, the Met Office and ECMWF have somewhat different experiences of the impact of coupled DA. It would be interesting to have a bit more of an idea why this may be. I know you can't explain the results of the other centres, but it would help to explore the mechanisms for the positive impacts you see.

Detailed comments

Page 1 line 12 abstract: "Next steps..." -> "The next steps..."

Page 3 line 5 I was confused by the sentence "The shorter synchronization time ..." It gives the impression the synchronization time is controllable whereas previously it is stated to be different in different parts of the ocean depending on the degree of cross correlation between the ocean and atmosphere. Please clarify the sentence and/or the paragraph. I wonder if changing this sentence to "The shorter synchronization time with strongly coupled DA ..." is correct.

Page 3 line 12. Perhaps: "... coupled reanalysis." -> "... coupled renalysis (described above)." Just to so it is clear for the reader that this is the system you summarised just above.

Page 5 line 2. "global 1/4 deg horizontal grid". Change "horizontal grid" -> "horizontal tripolar grid" or "horizontal ORCA grid" or similar.

Page 5 line 17. Include the name of the MDT. E.g. "The CNES-CLS09 mean dynamic topography is used from Rio et al (2011)".

Page 5 section 2.3. The SST assimilation method (in the ocean) seems less than ideal in that you are creating a gridded SST product with OI using the previous days gridded analysis as the background. This is then assimilated (again) into the ocean model (with the coupled model SST as the background). This means that where there are data gaps in effect the background of the gridded analysis is assimilated as observations in ocean (rather than retaining the ocean model background). Also the SST gridded product has correlations from the SST OI which should be accounted for when

these are assimilated into the ocean (which I don't believe is the case). For the uncoupled system this product is needed to provide the lower boundary condition for the atmosphere. And there is some benefit in the uncoupled framework in using the same SST in the ocean to keep the systems close together. Is this approach then somewhat of legacy and the method of SST assimilation will change in future when coupled DA becomes the standard? Can you add some discussion about these points to the text?

Page 6 line 13. Please expand a little on the practical reasons why the sea ice DA doesn't use a forecast model.

Page 7 section 3.1. I'm not sure I fully grasp how the weekly SAM2 analysis works within the daily cycling system. How are innovations calculated? How are the weekly SAM2 increments applied to the model? Over what time period and when? Presumably you don't rerun the whole week to apply the increments using IAU. How are the weekly increments combined with the increments from daily SST assimilation?

Page 8 lines 10-16. Can you clarify why it is necessary to save the precomputed atmospheric forcing fields in the ocean model? Is it to allow different ocean and atmosphere time windows?

Page 10 line 31. In the uncoupled experiment the same gridded SST is used in the ocean and atmosphere so I wonder why it is necessarily the case there is better accordance between the near-surface atmosphere temperature and the SST in the coupled experiment. Perhaps it is to do short time scale and diurnal variations - if so state that explicitly.

Page 11 line 20. There are quite big differences in the ocean by doing coupled DA. It is not really clear how this is arises since the are not many differences in the ocean DA same observations and SST product. Perhaps it is coming from changes in the atmosphere. I think this is an important result of the work and it merits some exploration of the reasons for the differences.

[Figure]

Page 18. Figure 1. Are the last three forcing states a model forecast or persistance?

Pages 21/22. Figures 4 and 5. What creates the saw tooth structure in the particularly the mean OmA

Pages 21/22. Figures 4 and 5. The caption UNCPL/CPL is hiding some the results. Please widen the time range so this is not the case.

Pages 21/22. Figures 4 and 5. Do you think there is overfitting of the data by the assimilation the Std dev doubles between the analysis and forecast?

Page 23. Figure 6. This plot would look better as a block/binned type plot as the contours are distracting when plotting binned data. (e.g for matplotlib use pcolor with interpolation = nearest).

Page 23. Figure 6. "Numbers show the areas .... statistically significant with confidence level above 90%". This also might work better with a block plot as the numbers could be in the centre of each block. I only see two numbers near the bottom left. Does this mean most of the plot is statistically insignificant? If that is the case is it worth showing?

Page 24. Figure 7. Consider adding a title "air temperatrue at 1000 hPa". You do have this for other figures and it makes it easier for the reader.

Page 24. Figure 7. Consider changing "the change in the " -> "the difference in the "

Page 25. Figure 8. Change the time axis so the dates do not run together (particular for the bottom panels).

Page 25. Figure 8. Change [60degS, 20degS] -> [60degS–20degS] etc. (and also on page 10).

Page 25. Figure 8. I see that the mean OmF is generally lower for UNCPL. You don't really comment on this in the text much. Expand the discussion on this a bit. Is it something to worry about? (The differences are small). And if not why? Also in the text

(page 10) you describe it as a bias (try to be consistent with the terminology used).

Page 25. Figure 9. Change y range for the bias plots. The red line is going off the top of the plot.

Page 25. Figure 9. There seems to a strong daily cycle in some of locations. Can you explain this? It's not discussed when you discuss this figure in the text.

Page 25/26. Figure 10/11. You have swapped the order of the mean and std deviation for these figures compared to the previous figures. Can you keep them in the same order (std dev first and mean/bias second)? It would help the reader also to always use the same linestyles as Fig 7 i.e dot-dashed lines for mean/bias even when they are in a separate panel.

Page 28. Figure 12. Change the colour range. It's a bit hard to see any details in the plot (at least when printed).

Page 28. Figure 12. Typo "W/m2"

Page 29/30. Figures 13/14. The areas described as grey in the caption come out blue in the plot. It would be better to replot so they are grey to distinguish better from the data.

Page 29/30. Figures 13/14/ The "Latitude" label is a bit close to the colour bar below.

Page 29/30. Figures 13/14. The section plots show quite a striking positive result for coupled DA. I would ask that you look a bit closer at the reasons for this good result for the coupled DA as otherwise it is not clear whether this a robust result.

---

## Referee Comment (RC2) · Anonymous Referee #2 · 24 Sep 2019

This paper describes the weakly coupled approach to data assimilation taken by ECCC for the initialisation of coupled NWP forecasts. The methodology is described in detail, and in particular compared to the uncoupled approach which is used as a control for assessment. Impact is measured by looking at observation statistics (analysis and forecast) and forecast error statistics.

The paper is overall of high quality, well written, and of scientific interest. There are no insurmountable problems with this manuscript, but I have one request which may be regarded as major, though I hope this could be addressed quickly.

Major point: The choice of forecast error statistic is one that I have not seen before

and I have had long discussions with colleagues about its applicability. Specifically it is the choice of verifying forecasts against the mean analysis of the two experiments. Starting with an example, if forecast 1 matches exactly analysis 1, then it will verify worse than if the forecast 2 drifts towards analysis 1, as it will approach the mean of analysis 1 and 2.

Looking in more detail, say at Simmons and Hollingsworth, 2002 (https://doi.org/10.1256/003590002321042135), differencing the equation for forecast errors on pg 668, you see that for this to be a true measure of forecast error differences then you are assuming that the true forecast errors times the correlation between the true forecast error and the mean analysis is equal for both forecasts ($f_{1T}c_{f_1a} = f_{2T}c_{f_2a}$). So then you have the additional problem of looking at how forecast 1 correlates with analysis 2, and vice versa. I think this just muddies the waters here!

The clean solution to this is to use an independent analysis for verification. Indeed, page 9 line 29 says that you have already produced such plots using ERA5. I would suggest to replace the results you show with those verified against ERA5 to simplify the interpretation of your results.

Minor points: Page 2, line 4: worth referencing ECMWF here: P. Bauer and D. Richardson. New model cycle 40r1. ECMWF Newsletter No. 138 - Winter 2013/2014, (138):3, 2014. URL https://www.ecmwf.int/node/14581

Page 4, line 14: are the increments computed on the full 80 levels? please clarify.

Page 7. line 26: "The daily ocean SAM2 DA (Sect. 2.2) assimilating only SST data is computed at 0000 UTC." Is this a daily mean SST field, or is it valid at 0000? Please clarify.

Page 8, line 10. Please could you clarify if the ensemble used in the 4D-EnVar uses a coupled or uncoupled model?

Page 8, line 12:14. "However, by saving the atmospheric fields from the 6-h coupled

forecasts and using these to force the ocean model, this is equivalent to the explicit use of the fully coupled atmosphere-ocean-ice model." My understanding of this line is as follows: "However, by saving the atmospheric fields from the 6-h coupled forecasts and using these to force the ocean model, this is equivalent to the explicit use of the fully coupled atmosphere-ocean-ice model with a 6 hour coupling frequency". Is this a correct reading? If so, should it be added for clarity?

Page 9, line 2: The test period used here of 2 months is short. Specifically it might be too short to see any major changes in the ocean component. Given the computational cost of the coupled assimilation experiments it would be unreasonable for anyone to ask for an extended period of testing. I think, however, this warrants a comment in the conclusion to reflect that the results should be viewed in this context.

Page 9, line 6:7. "Differences between these two systems are expected for the SST as well as for near-surface layers in both atmosphere and ocean models." This sentence I spent a while trying to understand what may be very obvious to the authors, and in the end I cannot see why SST is expected to be different in the two systems. I thought the SST analysis described in 2.3 was independent of any model, and so should not be different in the analysis of the weakly coupled or uncoupled systems. Perhaps this refers to forecasts of SST? Please can you expand on this to make your point more explicitly.

Page 10, line 7:8. "The OmF standard deviations produced by CPL are systematically lower than those produced by UNCPL in all three regions." This may be systematic, but it a is very small difference.

Page 12, line 24: "an integrated software" -> "integrated software"

Figures 6, 7 (top), 9, 10, 11: please state in the caption and in the text that these are plots of errors, not just std etc.

Figures 13 and 14: The grey colour looks blue which is misleading. Maybe replace the

orographic shading with a constant colour and make the grey areas that same colour.

---

## Author Comment (AC1) · 21 Oct 2019

**Response to Referee 2**

Sergey Skachko

October 2019

**We wish to thank the referee for the constructive review. Here below are our answers to the referee's questions and comments, in bold blue.**

This paper describes the weakly coupled approach to data assimilation taken by ECCC for the initialisation of coupled NWP forecasts. The methodology is described in detail, and in particular compared to the uncoupled approach which is used as a control for assessment. Impact is measured by looking at observation statistics (analysis and forecast) and forecast error statistics. The paper is overall of high quality, well written, and of scientific interest. There are no insurmountable problems with this manuscript, but I have one request which may be regarded as major, though I hope this could be addressed quickly.

Major point: The choice of forecast error statistic is one that I have not seen before and I have had long discussions with colleagues about its applicability. Specifically it is the choice of verifying forecasts against the mean analysis of the two experiments. Starting with an example, if forecast 1 matches exactly analysis 1, then it will verify worse than if the forecast 2 drifts towards analysis 1, as it will approach the mean of analysis 1 and 2. Looking in more detail, say at Simmons and Hollingsworth, 2002 ( https://doi.org/10.1256/003590002321042135 ), differencing the equation for forecast errors on pg 668, you see that for this to be a true measure of forecast error differences then you are assuming that the true forecast errors times the correlation between the true forecast error and the mean analysis is equal for both forecasts (f1T cf1a = f2T cf2a). So then you have the additional problem of looking at how forecast 1 correlates with analysis 2, and vice versa. I think this just muddies the waters here! The clean solution to this is to use an independent analysis for verification. Indeed, page 9 line 29 says that you have already produced such plots using ERA5. I would suggest to replace the results you show with those verified against ERA5 to simplify the interpretation of your results.

**The verification against ERA5 is shown on Fig. 1. As you can see, the results are similar to those obtained using the mean analysis, except the fact that the comparison against ERA5 doesn't show areas where the difference between two experiments is statistically significant with confidence level above 90%. On the other hand, the ERA5 reanalysis is not performed in a coupled data assimilation framework. Thus, comparing a coupled analysis with an uncoupled analysis with**

[Figure]

Figure 1: Difference in standard deviation of the air temperature, in degrees C, against the ERA5 reanalysis as a function of forecast lead time. The statistics are computed for CPL and UNCPL in the Northern Extratropics region. Positive values (red) mean that the standard deviation produced by CPL is smaller, whereas negative values (blue) mean the converse.

**respect to another uncoupled analysis may lead to preferencing the uncoupled analysis. Besides, the areas where the biggest statistically significant differences in standard deviation computed against the mean analysis are observed near the atmospheric-ocean interface, where the impact of coupling is expected. That is why we would prefer to keep the initial figure in the final manuscript.**

Minor points: Page 2, line 4: worth referencing ECMWF here: P. Bauer and D. Richardson. New model cycle 40r1. ECMWF Newsletter No. 138 - Winter 2013/2014, (138):3, 2014. URL https://www.ecmwf.int/node/14581

**Done.**

Page 4, line 14: are the increments computed on the full 80 levels? please clarify.

**The sentence is modified as follows: "The 4D-EnVar analysis increments are computed on a grid with a horizontal grid spacing of 50 km, as in the EnKF system, on all 80 vertical levels."**

Page 7. line 26: "The daily ocean SAM2 DA (Sect. 2.2) assimilating only SST data is computed at 0000 UTC." Is this a daily mean SST field, or is it valid at 0000? Please clarify.

**The sentence is modified as follows: "The daily ocean SAM2 DA (Sect. 2.2) assimilating only SST daily mean data is computed at 0000 UTC."**

Page 8, line 10. Please could you clarify if the ensemble used in the 4D-EnVar uses a coupled or uncoupled model?

**To do so, we modified the section 2.1 describing the atmospheric data assimilation: "The ensemble covariances are estimated from the ensemble of 256 uncoupled background states, available hourly within the 6-h assimilation window, obtained from the global ensemble Kalman filter (EnKF) being used operationally at ECCC (Houtekamer et al., 2014) since 2005."**

Page 8, line 12:14. "However, by saving the atmospheric fields from the 6-h coupled forecasts and using these to force the ocean model, this is equivalent to the explicit use of the fully coupled atmosphere-ocean-ice model." My understanding of this line is as follows: "However, by saving the atmospheric fields from the 6-h coupled forecasts and using these to force the ocean model, this is equivalent to the explicit use of the fully coupled atmosphere-ocean-ice model with a 6 hour coupling frequency". Is this a correct reading? If so, should it be added for clarity?

**The coupling frequency chosen for the computation of the CPL ocean analyses is one hour, as explained in the next sentence: "Preliminary experiments showed that the use of the precomputed atmospheric forcing from the fully coupled model every hour gives results similar to the forcing changing every model time step (the ocean-ice model time step is 15 min in our experiments)."**

Page 9, line 2: The test period used here of 2 months is short. Specifically it might be too short to see any major changes in the ocean component. Given the computational cost of the coupled assimilation experiments it would be unreasonable for anyone to ask for an extended period of testing. I think, however, this warrants a comment in the conclusion to reflect that the results should be viewed in this context.

**Following the referee's suggestion the following sentences are added to the conclusions: "Besides, the test period of two months used here might be too short to see major changes in the ocean component. Thus, the results presented here should be viewed in this context."**

Page 9, line 6:7. "Differences between these two systems are expected for the SST as well as for near-surface layers in both atmosphere and ocean models." This sentence I spent a while trying to understand what may be very obvious to the authors, and in the end I cannot see why SST is expected to be different in the two systems. I thought the SST analysis described in 2.3 was independent of any model, and so should not be different in the analysis of the weakly coupled or uncoupled systems. Perhaps this refers to forecasts of SST? Please can you expand on this to make your point more explicitly.

**The sentence is modified as follows: "Differences between these two systems are expected for the forecasts of SST and near-surface layers in both atmosphere and ocean models."**

Page 10, line 7:8. "The OmF standard deviations produced by CPL are systematically lower than those produced by UNCPL in all three regions." This may be systematic, but it a is very small difference.

**The text is modified as follows: "The OmF standard deviations produced by CPL are systematically lower than those produced by UNCPL in all three regions. Though, the differences are small."**

Page 12, line 24: "an integrated software" -¿ "integrated software"

**Done.**

Figures 6, 7 (top), 9, 10, 11: please state in the caption and in the text that these are plots of errors, not just std etc.

**Done.**

Figures 13 and 14: The grey colour looks blue which is misleading. Maybe replace the orographic shading with a constant colour and make the grey areas that same colour.

**Done.**

---

## Author Comment (AC2) · 25 Oct 2019

**Response to Anonymous Referee 1**

Sergey Skachko

October 2019

**We wish to thank the referee for the constructive review. Here below are our answers to the referee's questions and comments, in bold blue.**

Overview

This paper describes a development to the Canadian global prediction system at ECCC to use weakly coupled ocean atmosphere data assimilation. The impact of this is assessed against a system using the previous uncoupled ocean and atmosphere DA approach. The paper is overall well written and it will be valuable to publish the work. I do have a main comment and some detailed comments which I believe should be addressed first listed below.

Main comment

The are some positive impacts from the coupled DA but I didn't feel I understood why this was the case. You, the Met Office and ECMWF have somewhat different experiences of the impact of coupled DA. It would be interesting to have a bit more of an idea why this may be. I know you can't explain the results of the other centres, but it would help to explore the mechanisms for the positive impacts you see.

**There could be multiple reasons for the different impacts of WCDA starting from the most obvious, such as differences in coupling strategies, models, data assimilation methods and assimilated observations. However, what is clearly different in our system with respect to others is the ocean data assimilation system that uses two DA cycles, one with a weekly assimilation time window and the other with a daily window. As it is stated in the paper, the weekly DA cycle is not affected by the coupling process and therefore will prevent the deep ocean state from diverging between the UNCPL and CPL systems. In fact, every seven days, the ocean is restarted from the same uncoupled initial conditions in both UNCPL and CPL systems. This certainly affects the results, especially in the ocean, keeping it relatively close to the uncoupled solution. The small positive impact of WCDA on the ocean forecast that we observe may be likely attributed to the better consistency between ocean and atmosphere when using the coupled analysis, which reduces the initial shocks in the coupled forecasts. This short discussion is added to the text.**

Detailed comments

Page 1 line 12 abstract: "Next steps..." - "The next steps..."
**Done.**

Page 3 line 5 I was confused by the sentence "The shorter synchronization time ..." It gives the impression the synchronization time is controllable whereas previously it is stated to be different in different parts of the ocean depending on the degree of cross correlation between the ocean and atmosphere. Please clarify the sentence and/or the paragraph. I wonder if changing this sentence to "The shorter synchronization time with strongly coupled DA ..." is correct.

**The sentence is changed as you proposed: "...a shorter synchronization time with strongly coupled DA."**

Page 3 line 12. Perhaps: "... coupled reanalysis." - "... coupled renalysis (described above)." Just to so it is clear for the reader that this is the system you summarised just above.

**Done.**

Page 5 line 2. "global 1/4 deg horizontal grid". Change "horizontal grid" -¿ "horizontal tripolar grid" or "horizontal ORCA grid" or similar.

**Done.**

Page 5 line 17. Include the name of the MDT. E.g. "The CNES-CLS09 mean dynamic topography is used from Rio et al (2011)".

**Done.**

Page 5 section 2.3. The SST assimilation method (in the ocean) seems less than ideal in that you are creating a gridded SST product with OI using the previous days gridded analysis as the background. This is then assimilated (again) into the ocean model (with the coupled model SST as the background). This means that where there are data gaps in effect the background of the gridded analysis is assimilated as observations in ocean (rather than retaining the ocean model background). Also the SST gridded product has correlations from the SST OI which should be accounted for when these are assimilated into the ocean (which I don't believe is the case). For the uncoupled system this product is needed to provide the lower boundary condition for the atmosphere. And there is some benefit in the uncoupled framework in using the same SST in the ocean to keep the systems close together. Is this approach then somewhat of legacy and the method of SST assimilation will change in future when coupled DA becomes the standard? Can you add some discussion about these points to the text?

**As you point out, this approach is indeed a legacy of the uncoupled data assimilation approach. It is designed to reduce initialization shocks for coupled forecasts by imposing consistent surface conditions on the atmosphere and ocean during their separate assimilation cycles. The assimilation of an SST OI analysis by an ocean analysis system is also fairly commonplace within the operational oceanographic community (e.g. by the Global Ocean Forecasting system within the Copernicus Marine Environmental Monitoring System, Lellouche et al., 2013. This approach has the benefit that it provides a pre-treatment (or superobbing) of high-resolution SST observations allowing the assimilation of more satellite SST datasets. Since the**

**coverage oImplement and test a stand-alone sea surface temperature (SST) analysis within the MIDAS code framework used for atmospheric data assimilation.f satellite SST data is quite complete, data gaps don't produce a significant problem compared to the advantages of this approach. A brief discussion of this point has been added to the text.**

Page 6 line 13. Please expand a little on the practical reasons why the sea ice DA doesn't use a forecast model.

**Similar to the SST analysis, the sea ice analysis system was originally developed using various sources of satellite data that generally have good combined spatial coverage globally every 6h. This system was developed before our center became active with using sea-ice models and therefore it was for practical reasons that it does not use a model within the assimilation cycle. In the future, like for SST, we plan to use the coupled ice-ocean-atmosphere model to provide the background state when exploring strongly coupled data assimilation. These sentences are added to the text.**

Page 7 section 3.1. I'm not sure I fully grasp how the weekly SAM2 analysis works within the daily cycling system. How are innovations calculated? How are the weekly SAM2 increments applied to the model? Over what time period and when? Presumably you don't rerun the whole week to apply the increments using IAU. How are the weekly increments combined with the increments from daily SST assimilation?

**Each Wednesday, two consecutive 7-day analysis cycles are produced (e.g. covering the previous 14 days). For each 7-day cycle, a trial run is first produced computing innovations using the First Guess at the Appropriate Time (FGAT) approach. The model is then reinitialized from day 6 of the cycle and a 1-day IAU run is made over day 7. The daily cycling system is then initialized from the weekly analysis each Wednesday, and continued throughout the week. Further details of the method can be found in Smith et al 2015 and http://collaboration.cmc.ec.gc.ca/cmc/cmoi/product_ guide/docs/tech_notes/technote_giops-210_e.pdf.**

Page 8 lines 10-16. Can you clarify why it is necessary to save the precomputed atmospheric forcing fields in the ocean model? Is it to allow different ocean and atmosphere time windows?

**The current ocean data assimilation is a complex Mercator SAM2 system installed on the ECCC's supercomputers and run operationally. The system is built on the NEMO model coupled with a data assimilation scheme. Using the precomputed atmospheric forcing fields from the coupled ocean-atmosphere forecasts allows us to compute implicitly coupled ocean background states without modifying the SAM2 code.**

Page 10 line 31. In the uncoupled experiment the same gridded SST is used in the ocean and atmosphere so I wonder why it is necessarily the case there is better accordance between the near-surface atmosphere temperature and the

SST in the coupled experiment. Perhaps it is to do short time scale and diurnal variations - if so state that explicitly.

**While the same gridded SST OI analysis is assimilated in the ocean and used for the surface boundary condition in UNCPL, differences remain between the SST of the ocean analysis and the OI analysis. The OI analysis is assimilated with a $0.3°C$ observation error by the ocean analysis and thus differences of up to $1.0°C$ can be found. These differences are especially apparent in energetically active areas of the ocean (due to small scale eddies not captured in the OI analysis) as well as due to the presence of cyclones (e.g. due to cold wakes). As a result, surface fluxes in UNCPL forecasts will reflect this imbalance in initial conditions. The text has been modified to clarify this point.**

Page 11 line 20. There are quite big differences in the ocean by doing coupled DA. It is not really clear how this is arises since the are not many differences in the ocean DA same observations and SST product. Perhaps it is coming from changes in the atmosphere. I think this is an important result of the work and it merits some exploration of the reasons for the differences.

**The section plots are shown here only to illustrate possible impact of WCDA on the ocean vertical structure. These plots are made in the areas where the difference between CPL and UNCPL reaches maximum values near the surface. In these areas the mean difference in the ocean temperature is of order 0.1 C. Overall, the impact of the WCDA on the vertical ocean structure is rather neutral as stated in the text.**

Page 18. Figure 1. Are the last three forcing states a model forecast or persistance?

**These states are obtained from the coupled model forecast.**

Pages 21/22. Figures 4 and 5. What creates the saw tooth structure in the particularly the mean OmA

**For the mean, we see clearly a 24-h period, so it is likely related to diurnal cycle. However, for the STD, it looks like a 12h variation, so not due to diurnal cycle. This effect is probably related to the data coverage.**

Pages 21/22. Figures 4 and 5. The caption UNCPL/CPL is hiding some the results. Please widen the time range so this is not the case.

**Done.**

Pages 21/22. Figures 4 and 5. Do you think there is overfitting of the data by the assimilation the Std dev doubles between the analysis and forecast?

**No, we don't think so. First of all, comparing the OmA/OmF of different satellite radiance instruments, that are assimilated in our NWP system, we see similar behaviour in different channels, sensitive to the surface or not. These radiance observation errors were obtained by conducting extensive tuning experiments to determine the values providing the best forecast quality. In addition, the increment of the skin temperature, which is not negligible in this surface-sensitive channel, computed in our 4D-EnVar atmospheric data as-**

**similation system is not cycled in the current NWP system (neither in the WCDA). Our future work towards strongly coupled data assimilation will use this increment for initializing the coupled forecast and therefore it will be interesting to revisit at that time how these surface-sensitive channels are assimilated.**

Page 23. Figure 6. This plot would look better as a block/binned type plot as the contours are distracting when plotting binned data. (e.g for matplotlib use pcolor with interpolation = nearest).

**This figure comes from our automated NWP verification system. Therefore, to maintain consistency with previous publications, we prefer to keep the figure unchanged.**

Page 23. Figure 6. "Numbers show the areas .... statistically significant with confidence level above 90%". This also might work better with a block plot as the numbers could be in the centre of each block. I only see two numbers near the bottom left. Does this mean most of the plot is statistically insignificant? If that is the case is it worth showing?

**Yes, this means that the scores are statistically not significant above 90% confidence level elsewhere. We have modified the text to state that explicitly.**

Page 24. Figure 7. Consider adding a title "air temperatrue at 1000 hPa". You do have this for other figures and it makes it easier for the reader.

**Done.**

Page 24. Figure 7. Consider changing "the change in the " - "the difference in the "

**Done.**

Page 25. Figure 8. Change the time axis so the dates do not run together (particular for the bottom panels).

**Done.**

Page 25. Figure 8. Change [60degS, 20degS] - [60degS–20degS] etc. (and also on page 10).

**Done.**

Page 25. Figure 8. I see that the mean OmF is generally lower for UNCPL. You don't really comment on this in the text much. Expand the discussion on this a bit. Is it something to worry about? (The differences are small). And if not why? Also in the text (page 10) you describe it as a bias (try to be consistent with the terminology used).

**Indeed, the differences for the mean OmF are small and also not systematically in favor of one experiment over the other (as opposed to the difference for the standard deviation of OmF). Due to this variation in the differences for the mean OmF and the relatively short length of the experiments, we are not confident in the statistical significance of the differences. This is now briefly mentioned in the text.**

Page 25. Figure 9. Change y range for the bias plots. The red line is going off the top of the plot.

**Done.**

Page 25. Figure 9. There seems to a strong daily cycle in some of locations. Can you explain this? It's not discussed when you discuss this figure in the text.

**The coupled forecasts represent the diurnal cycle to some extent for the SST, which is not captured in the ERA5 SST analyses (since it is a foundation SST). Therefore for the Northern Extratropics the mean difference in SST has a strong daily cycle since the Pacific ocean dominates the oceanic surface area at these latitudes, which is mostly day at 0000 UTC and night at 1200 UTC. In contrast, the other regions have more even oceanic coverage at all longitudes (especially for the Southern Extratropics), and therefore have a more even coverage of day and night at both 0000 UTC and 1200 UTC. This is now briefly mentioned in the text.**

Page 25/26. Figure 10/11. You have swapped the order of the mean and std deviation for these figures compared to the previous figures. Can you keep them in the same order (std dev first and mean/bias second)? It would help the reader also to always use the same linestyles as Fig 7 i.e dot-dashed lines for mean/bias even when they are in a separate panel.

**Done.**

Page 28. Figure 12. Change the colour range. It's a bit hard to see any details in the plot (at least when printed).

**Done.**

Page 28. Figure 12. Typo "W/m2"

**Done.**

Page 29/30. Figures 13/14. The areas described as grey in the caption come out blue in the plot. It would be better to replot so they are grey to distinguish better from the data.

**Done.**

Page 29/30. Figures 13/14/ The "Latitude" label is a bit close to the colour bar below.

**Done.**

Page 29/30. Figures 13/14. The section plots show quite a striking positive result for coupled DA. I would ask that you look a bit closer at the reasons for this good result for the coupled DA as otherwise it is not clear whether this a robust result.

**See our response to the** comment for Page 11 line 20